# In-Silico Screening and Molecular Dynamics Simulation of Drug Bank Experimental Compounds against SARS-CoV-2

**DOI:** 10.3390/molecules27144391

**Published:** 2022-07-08

**Authors:** Norah A. Alturki, Mutaib M. Mashraqi, Ahmad Alzamami, Youssef S. Alghamdi, Afaf A. Alharthi, Saeed A. Asiri, Shaban Ahmad, Saleh Alshamrani

**Affiliations:** 1Clinical Laboratory Science Department, College of Applied Medical Science, King Saud University, Riyadh 11433, Saudi Arabia; noalturki@ksu.edu.sa; 2Department of Clinical Laboratory Sciences, College of Applied Medical Sciences, Najran University, Najran 61441, Saudi Arabia; mmmashraqi@nu.edu.sa (M.M.M.); saaasiri@nu.edu.sa (S.A.A.); 3Clinical Laboratory Science Department, College of Applied Medical Science, Shaqra University, Al Quwayiyah 11961, Saudi Arabia; aalzamami@su.edu.sa; 4Department of Biology, Turabah University College, Taif University, P.O. Box 11099, Taif 21944, Saudi Arabia; ysghamdi@tu.edu.sa; 5Department of Clinical Laboratory Sciences, College of Applied Medical Sciences, Taif University, P.O. Box 11099, Taif 21944, Saudi Arabia; a.awwadh@tu.edu.sa; 6Agriculture Knowledge Management Unit, ICAR-Indian Agricultural Research Institute, New Dehli 110012, India; shaban184343@st.jmi.ac.in

**Keywords:** SARS-CoV-2, RNA-dependent RNA polymerase, replication-transcription complex, molecular docking, molecular dynamics simulation

## Abstract

For the last few years, the world has been going through a difficult time, and the reason behind this is severe acute respiratory syndrome-coronavirus-2 (SARS-CoV-2), one of the significant members of the Coronaviridae family. The major research groups have shifted their focus towards finding a vaccine and drugs against SARS-CoV-2 to reduce the infection rate and save the life of human beings. Even the WHO has permitted using certain vaccines for an emergency attempt to cut the infection curve down. However, the virus has a great sense of mutation, and the vaccine’s effectiveness remains questionable. No natural medicine is available at the community level to cure the patients for now. In this study, we have screened the vast library of experimental drugs of Drug Bank with Schrodinger’s maestro by using three algorithms: high-throughput virtual screening (HTVS), standard precision, and extra precise docking followed by Molecular Mechanics/Generalized Born Surface Area (MMGBSA). We have identified 3-(7-diaminomethyl-naphthalen-2-YL)-propionic acid ethyl ester and Thymidine-5′-thiophosphate as potent inhibitors against the SARS-CoV-2, and both drugs performed impeccably and showed stability during the 100 ns molecular dynamics simulation. Both of the drugs are among the category of small molecules and have an acceptable range of ADME properties. They can be used after their validation in *in-vitro* and *in-vivo* conditions.

## 1. Introduction

The members of the *coronaviridae* family have caused various destruction earlier, and now one of its members has caused a global pandemic resulting in millions of deaths and a massive loss to various countries’ economic growth. This is all a result of severe acute respiratory syndrome-coronavirus-2 (SARS-CoV2) [1]. It has crown-like spikes on the surface and belongs to the Coronaviridae family. The species list is enormous in this family, and previously, it attracted attention due to infection of 2003 in SARS-CoV and Middle East respiratory syndrome (MERS) [2,3,4]. The novel strain of SARS-CoV causing COVID-19 shows mild to severe respiratory illness symptoms. With the newest variants, the symptoms may vary from fever, dry cough, sour throat, and breathing difficulties leading to death in the worst of cases [5,6]. The World Health Organization (WHO) has declared (COVID-19) a pandemic on 11 March 2020 [5]. By analyzing the reported data, 276,724,130 people were infected worldwide while 5,388,449 died by 22 December 2021, and uncountable are not even reported as the world still lacks the health infrastructure countries were not even prepared enough.

Most people infected with COVID-19 experience mild to moderate symptoms but recover without special treatment as their innate immunity works against the virus. Compared to the previous SARS-CoV and MERS outbreak, this virus remains for a more extended period. Instead, it becomes pandemic and spreads worldwide in just a few months, and the various strains have still been reported from different countries, and the uncertainties continue till data. The COVID-19 or SARS-CoV2 disease has a very high spread rate, which was the reason behind a huge distressing. The relaxation was there as it causes death with a less fatality rate (IFR) of 1.4% worldwide than other viral infections [7,8]. At the beginning of 2020, the SARS-CoV-2 genome was sequenced and made public, easing the researchers’ work to understand the SARS-CoV-2 viral system’s genomic component and develop diagnostic kits [9].

Moreover, from the same time, a few labs started crystallizing the viral proteins and a few labs crystallizing viral and human interacting complex (spike-ACE-2), and it helped another researcher to carry out the drug screening through repurposing models and novel candidates designed to alleviate the pandemic situation from the world [10]. After so much research, the mystery is not solved, and no potential drug candidate has been reported yet in the market that can treat the patients with the disease or boost the immunity to work against SARS-CoV-2. It is also essential to comprehend why this virus has a high infection rate and to determine how we have prospered for treatments against the pandemic we are going through. The proteomics data have helped scientists worldwide screen the vast library through molecular docking, pharmacophore modelling, and artificial intelligence and machine learning in drug designing to comprehend the process and make the drug available for human beings on at least an emergency basis. Even after a considerable loss and deaths of infected people, the uncertainty is there, and it continues as a result of various evolving variants. Despite the severity of the pandemic, there is no effective therapeutic available on the market. Scientists are working on drug repurposing or designing a novel candidate against SARS-CoV-2 to slow down infection and fatalities and take the world for positive development [1].

In this study, we have taken the experimental library of the drug bank database to screen against two potential drug targets that can produce a better result to assist in medication development. Further, our study extended to molecular dynamics simulation and its extensive analysis to understand the interacting residues properly.

## 2. Methodology

The methodology of the complete study contains a few sets of the section, and we have plotted a graphical abstract in Figure 1. The detailed methods description of the methods is as follows:

### 2.1. Protein Preparation

The structure of severe acute respiratory syndrome-coronavirus-2 RNA-dependent RNA polymerase (PDB ID: 6M71) and replication-transcription complex (PDB ID: 6XEZ) downloaded from the RCSB database has no mutations, and the resolution of 2.90 Å and 3.50 Å, respectively. The originally downloaded PDB files do not have proper bonding configuration, neither the hydrogen atoms are satisfied to further use for any studies. The proteins were prepared using the ‘protein preparation wizard’ from Schrodinger Maestro (https://www.schrodinger.com/) (V.12.8.117) to fix all problems and generate and ready to dock protein. Bonding orders were allocated with the CCD database, and corresponding hydrogen atoms were added to optimize the process. The prime module fills the missing loops and side chains in the same wizard. The hetero state was generated using Epik with a pH of 7.0, and zero bond orders were kept to create the disulphide bonds further [11]. There were three chains, A, B, C and D, only chain A kept in the RNA-dependent RNA polymerase, and all other dimers and short chains were removed. In contrast, chain A was kept after review in the replication-transcription complex case, and other solvents and ligands were removed. Further, using the refine tab, we have optimized the H-bonds to fix all the problems in protein and minimized them with the OPLS_2005 forcefield after removing the water molecules beyond 3.0Å of heteroatoms [12].

### 2.2. Ligand Library Collection and Preparation

The Drug Bank id was created and approved by the Drug Bank team to access the data after agreeing to use it for scientific purposes [13,14]. After logging in to the database, drugs belonging to the experimental category were downloaded (6658 drugs) and imported to the workspace. Further, the LigPrep wizard was used to prepare the ligands. Some were initially in 2-D format and not ready to dock even though the hydrogen atoms did not satisfy the valency criteria [14,15]. We selected 7 pH with (±2) to generate the best possible states using the Epik module and generated the tautomers, and at most, 32 stereoisomers were kept for each ligand.

### 2.3. Active Site Calculation and Glide Grid Generation

The active site of the proteins (6M71 and 6XEZ) was computed using the SiteMap module in the Schrodinger suite [16]. To compute the active site, in the setting, identify top-ranked potential receptor binding sites and crop site maps at 4 Å from the nearest site point kept that generated a total of five active sites. The ‘receptor grid generation’ wizard generated the grid file on the active site. For both cases, site1 was used to calculate the grid around, and the box size was increased to fit the complete active site and further performed molecular docking on the same grid file [17].

### 2.4. Molecular Docking and ADMET Analysis

The virtual screening workflow (VSW) is used to perform molecular docking, with all three algorithms to screen and score. The high throughput virtual screening (HTVS) with 90%, standard precision (SP) with 90%, and extra precise (XP) with 100% candidates carried out to post-processed with MMGBSA [18,19,20]. The ligands were filtered against their ADMET properties using the QikProp module and refined with Lipinski’s rule based on ADMET properties further, and the ligands were regularized with input geometry and duplicates were removed. Further, the generated grid was added to run the molecular docking selecting all three algorithms [21]. Empirical scoring of the Glide can be understood by:***GlideScore = c × E_coul_ + c × E_vdw_ + c × E_lipo_+ c × E_Hbond_ + c × E_metal_ + c × E_rotb_ + c × E_polar_phob_ +c × E_rewards_***(1)

Further, the compounds’ absorption, distribution, metabolism, excretion, and toxicity (ADMET) were analyzed using the QikProp tool in the maestro and filtered with Lipinski’s rule of 5 for further assessment [22,23].

### 2.5. Molecular Dynamics Simulation

After analyzing the ligand interaction diagram, only the top one complex (docking score) from each docking parameter was taken for the molecular dynamics (MD) simulation. Desmond package in Schrödinger suite v2021-3 was used to run the MD simulation to elucidate the effectiveness of the screened compounds by molecular docking [24]. The ‘system builder’ was used to prepare the protein-ligand complex. The SPC water model in an orthorhombic shape was selected after minimizing the volume, with 10 Å × 10 Å × 10 Å periodic boundary conditions in the P-L complex’s x, y, and *z*-axis. Moreover, 4Na^+^ was added to RNA-dependent RNA polymerase, and 13Na^+^ was added to replication-transcription to neutralize the system. Ion and salt placement within 20 Å were excluded from making the simulation neutralized.

Further, using the OPLS2005 forcefield, the complex minimized its energies by heating and equilibrium processes before the production run of MD simulations [12]. The steepest descent method-based minimization protocol was used against the complexes, then heated at 0–300 K. Further, with the time step of 100 ps, the system normalized in an equilibrium state at 1000 steps. The final production run was kept for 100 ns, at the time steps of 100 ps, 300 K temperature and 1.01325 atm pressure, for both complexes applying the Nose-Hoover method with NPT ensemble [25].

## 3. Results

### 3.1. Molecular Docking

The virtual screening workflow has given a total of 691 docked ligands which were further filtered on behalf of the docking score and MMGBSA score and analyzed the docking score and its interacting residues. Additionally, we have only taken protein-ligand complex (RNA-dependent RNA polymerase (6M71) with 3-(7-diaminomethyl-naphthalen-2-YL)-propionic acid ethyl ester and replication-transcription complex (6XEZ) with Thymidine-5′-thiophosphate) for further molecular dynamics simulation [26,27]. The protein-ligand interaction of the stable docked RNA-dependent RNA polymerase (6M71) with 3-(7-diaminomethyl-naphthalen-2-YL)-propionic acid ethyl ester and replication-transcription complex (6XEZ) with Thymidine-5′-thiophosphate complex was visualized with the visualizer tool named ligand-interaction diagram to analyze the interacting residues properly. The 3-(7-diaminomethyl-naphthalen-2-YL)-propionic acid ethyl ester shows a few interactions with the amino acids, i.e., NH_2_^+^ created one pi-cation with TYR129 also the same NH_2_^+^ shows one hydrogen bond with ASP126. NH_2_^+^ and NH_3_^+^ establish salt bridges with ASP126 and ASP208 (Figure 2A). The Thymidine-5′-thiophosphate shows different bonding configurations; the phosphate group (of O^−^) creates four salt bridges with ARG33, ARG55, ARG55, and LYS50. While the LYS73, ASN209, LYS50, and ASP208 created hydrogen bonds with oxygen and hydroxy group (Figure 2B). The importance of the above drugs in repurposing gives a huge potential as it is already under investigation with other diseases and public utilization for asthma-related problems. Our in-silico screening through different algorithms of molecular docking studies and the prime-MM-GBSA results showed an encouraging output for confirming the compound’s activity. It predicts that the drugs can be further tested in in-vitro and in-vivo labs to understand the exciting activity against the viral proteins of SARS-CoV-2. The Prime MM-GBSA produced the binding free energy of the complex, shown in Table 1.

Further, we have analyzed the ADMET properties of both the ligands 3-(7-diaminomethyl-naphthalen-2-yl)-propionic acid ethyl ester (DB07639) and thymidine-5′-thiophosphate (DB08432). Table 2 has provided the complete details of the standard properties and drugs’ properties. All the properties match the criteria of the standard drug candidate, and the oral absorption of both the ligands is good. It can be provided as an oral drug, and its absorption percentage is also good, meaning the drug will be efficiently absorbed and excreted from the human body after its effect against the disease.

### 3.2. Molecular Dynamics Simulation

The molecular dynamics simulation is a method and set of algorithms that calculate and predict compounds’ stability. It is one of the best stands alone mechanisms for the fundamental computational tool to capture the molecular and atomistic-level changes and is essential to know the stability of protein-ligand complex (through deviation and fluctuation analysis) and intermolecular interactions. The structure-based drug design using the conventional approach such as molecular docking and virtual screening provided the shortlisted drugs in bioscience. At the same time, the MD simulation plays an essential role in understanding the ligand’s dynamic behavior and its stability against the protein. For 100 ns SPC water model-based simulation, we analyzed the MD simulation trajectories with simulation interaction diagram (SID) to understand the deviation, fluctuation and intermolecular interaction.

#### 3.2.1. RMSD and RMSF

Root mean square deviation (or RMSD) value is used to calculate deviation in the backbone of the protein (Cα, C, and N) during the 100 ns of the simulative period. The complex initially fluctuates when the temperature goes up and then stabilizes. Both proteins did not deviate much during the entire simulation period. In the condition of RNA-dependent polymerase in a complex with 3-(7-diaminomethyl-naphthalen-2-YL)-propionic acid ethyl ester, the protein RMSD initially fluctuated from 0 to 1.63 Å in 0.50 ns, and the ligand fluctuated 2.51 Å (Figure 3A). While in the condition of Replication-transcription complex in complex with Thymidine-5′-thiophosphate in the 0.2 ns, the protein fluctuated to 1.99 Å, and the ligand fluctuated to 3.43 Å (Figure 3B). The overall RMSD is satisfactory for both combinations. After initial fluctuation, it stabilized for the complete duration of the simulation, and the fundamental fluctuations are due to the initial heat solution medium for the whole complex. The protein of RNA-dependent polymerase in complex with 3-(7-diaminomethyl-naphthalen-2-YL)-propionic acid ethyl ester shows the deviation of 2.78 Å while the ligand shows 2.56 Å at 100 ns. The protein can be defined under 1.15 Å of deviation after trimming the initial 0.50 ns, while for the ligand, 0.05 Å after the exact initial trimming. It means that the complex is completely stable for the 100 ns simulation. At 100 ns, the protein of Replication-transcription in complex with Thymidine-5′-thiophosphate shows a deviation of 3.35 Å, and the ligand shows a 3.02 Å deviation. The protein can be defined under 1.36 Å of deviation after trimming the initial 0.20 ns, while the ligand 0.41 Å after the exact initial trimming. It means that the complex is completely stable for the 100 ns simulation.

Later, the root mean square fluctuation (or RMSF) analysis gives the complex variations with time evolution against each atom. In Figure 4A, we have shown the P-RMSF and the protein contacts with ligand for the complete simulation. We have demonstrated the RNA-dependent polymerase in complex with 3-(7-diaminomethyl-naphthalen-2-YL)-propionic acid ethyl ester concerning their contacts (protein-ligand) during 100 ns simulations.

In RNA-dependent polymerase (Figure 4B), ILE114, LEU270, GLU431, TYR595 and LEU895 have fluctuated. The rest of the residues have shown a significantly less acceptable level of fluctuations, and 3-(7-diaminomethyl-naphthalen-2-YL)-propionic acid ethyl ester has shown 22 times contact with the protein. While among the all-amino acids of Replication-transcription VAL14, ASP336, LYS391, ARG392, SER904, and THR929 have demonstrated the most fluctuation with 21-time contact with Thymidine-5′-thiophosphate during the complete 100 ns simulations. Further, the overall noticed fluctuation is very low, giving a piece of enormous information to use both drugs for further studies against CoV-2. Furthermore, the intermolecular interactions (H-bond, pi-pi stacking) and secondary structure elements (alpha helices and beta strands) make the protein molecule lightly rigid. The fluctuation showed in Figure 4A,B clearly can be seen below to 2 Å in both conditions, showing promising results.

#### 3.2.2. Intermolecular Interaction

The atomic-level interaction information is important to predict the binding mode of the protein and ligand during the complete simulation process. Intermolecular interaction among protein and ligand molecules such as hydrogen bond, ionic interaction, hydrophobic contact, and the salt bridge was extensively analyzed for binding analysis during the complete 100 ns simulation.

This study proves that many intramolecular interactions are participating, such as hydrogen bonds, pi-pi stacking, and water molecules’ involvement in water bridges. Figure 5A shows 3-(7-diaminomethyl-naphthalen-2-YL)-propionic acid ethyl ester interaction with the amino acids of RNA-dependent polymerase and other relevant fragments. Even though we have not noticed any direct interaction with carbon molecules, the interaction with the NH_2_ and NH_3_^+^ groups formed the hydrophilic, hydrophobic and hydrogen bonding interactions with respective percentiles. Further, the direction of the arrows shows donors as well as the electron acceptors. The H_2_O molecules interacted widely, forming the water bridges, while the amino acids interacted directly as well as through hydrophilic and other interactions. There are six water molecules involved in interaction along with THR206, ASP208, TYR728, HIS133, LEU708, and ASP126 are, forming the hydrogen bond TYR728 forms two pi-pi stacking while TYR129 forms single pi-pi stacking with the shown respective percentile of interaction during 100 ns simulations.

Figure 5B shows the Thymidine-5′-thiophosphate interaction with the amino acids of RNA-dependent polymerase. Interestingly, there are two different bonding noticed: PHE35 forms pi-pi stacking with one benzene ring. Five water molecules are involved in this interaction, and ASP218, LYS50, ASN209, and LYS73 are with other ligand molecules while the TYR217, ARG33, ARG55, CYS53, ASN52 with the phosphate group of the ligand are forming hydrogen bonding interaction with different atoms of the ligand. Further, the statistical interpretations are made in Figure 6A,B, which indicate the H-bond count, hydrophobic interactions, ionic interactions, and water bridges.

## 4. Discussion

The SARS-CoV-2 virus has caused the death of millions of humans worldwide, and it still evolves itself in the form of different strains. The uncertainties continue to date even after the availability of the vaccines. However, there is a gap in a particular drug that can treat the patient and retain the virus [28]. Discovering a potent target and bringing it to the market will be the revolutionaries’ research of this decade. There is a desperate need for drug compounds to target the proteins of the SARS-CoV-2 virus. We have downloaded and prepared the complete experimental library of the Drug Bank database, which gives vast data, and we have prepared them to be ready to dock to the protein complexes [13]. Further, two main targets, RNA-dependent polymerase and replication-transcription targeted selected to screen the prepared library. The molecular docking gives us tremendous results, and the one complex from each docked development is taken for the MD simulation.

Interestingly, after the exhaustive screening, the 3-(7-diaminomethyl-naphthalen-2-YL)-propionic acid ethyl ester showed a docking score of −8.781 and MMGBSA score of 45.78 with RNA dependent-polymerase while the Thymidine-5′-thiophosphate) with Replication transcription has shown the docking score of −8.582 and MMGBSA score of 55.492. The molecular docking results fascinate us, and 3-(7-diaminomethyl-naphthalen-2-YL)-propionic acid ethyl ester (DB07639) and Thymidine-5′-thiophosphate (DB08432) these drugs are being experimentally validated for many other diseases. Both the simulated ligands belong to a small group of ligands. Further, the MD simulation results also showed the deviation and fluctuation in less than 2A with multiple contacts during the 100 ns of simulation. The mechanism of action of the proposed drugs is that they stimulate the Beta (2)-receptor in the lung, which causes relaxation of bronchial smooth muscle, bronchodilation, and increased bronchial airflow that provides haptic ease to the patients. Further, the efficacy of these candidates needs to be experimentally reverified before human use.

## 5. Conclusions

The reason behind selecting the Drug Bank database is that it provides the drugs in a well-categorized manner and interestingly, the database is well annotated among all drug databases. The drug bank provides a list of experimental drugs tested against multiple diseases. We have applied the three algorithms to reduce the computational cost as the HTVS takes almost 1-2 s/ligand, SP takes almost 1 min, and then applies to the XP docking algorithm that can take up to 3 min to give the most accurate results after extensive sampling. Although the molecular docking study has given promising results with many candidates, we only selected the top 1 ligand from both results to further simulate using the SPC water model. The results for both ligands, i.e., 3-(7-diaminomethyl-naphthalen-2-YL)-propionic acid ethyl ester and Thymidine-5′-thiophosphate, were produced a promising docking score, binding free energy (dG bind) and in the MD simulation results. The MMGBSA produces the binding free energy that shows how tightly drugs are bound to the protein.

Interestingly, after getting relevant results for MMGBSA analysis, we have also obtained promising results from molecular dynamics simulation with significantly less deviation and fluctuation, including an excellent binding state with hydrogen bond and pi-pi stacking. The produced results from all pipelines were promising and gave enough evidence that the said drugs can be potent against SARS-CoV-2. The potency of both studied compounds needs the next level of validation in the experimental condition of in vitro and in-vivo.

## Figures and Tables

**Figure 1 molecules-27-04391-f001:**
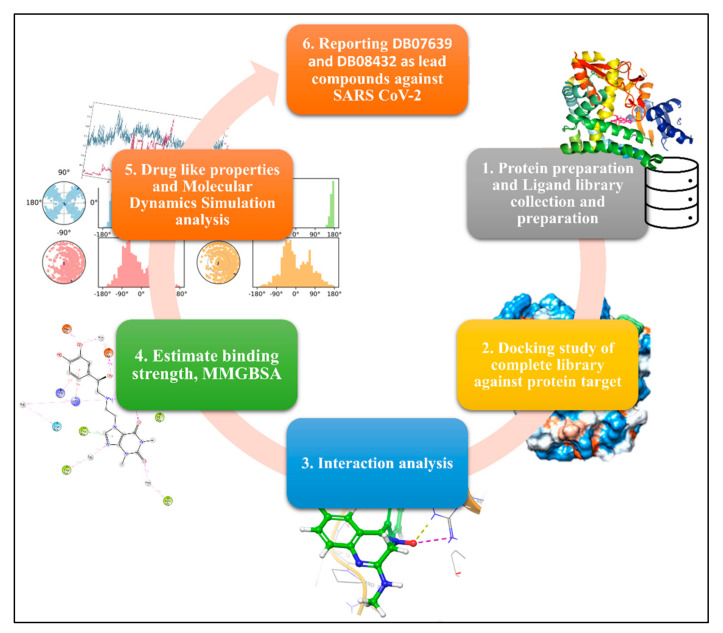
Showing the graphical abstract of the study; A workflow from protein and drug library collection to the molecular dynamics simulation.

**Figure 2 molecules-27-04391-f002:**
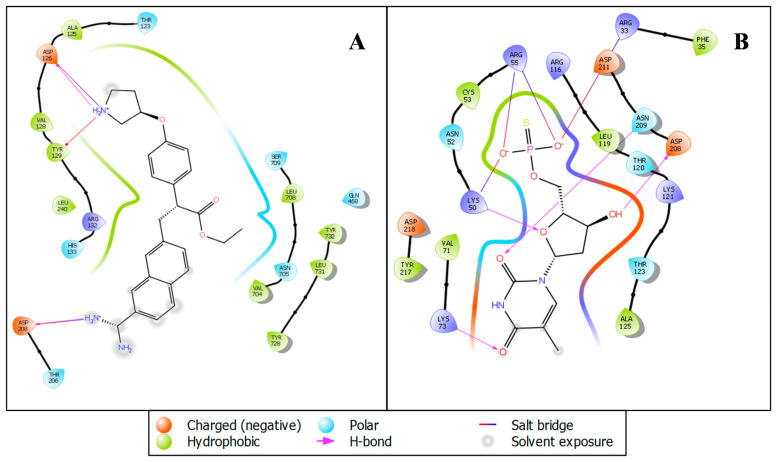
Ligand Interaction diagram of (**A**) RNA-dependent RNA polymerase (6M71) with 3-(7-diaminomethyl-naphthalen-2-YL)-propionic acid ethyl ester and (**B**) replication-transcription complex (6XEZ) with Thymidine-5′-thiophosphate showing interacting residues and interaction types.

**Figure 3 molecules-27-04391-f003:**
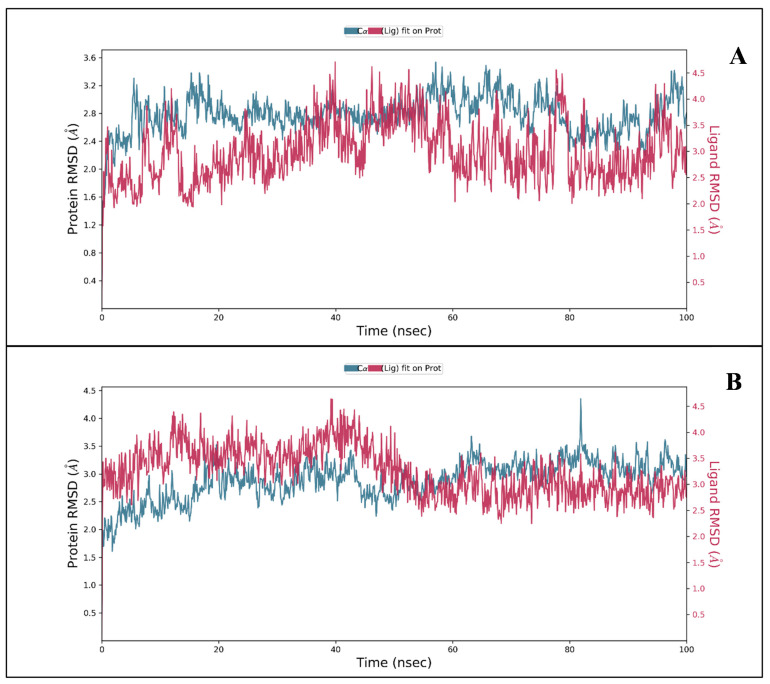
Showing the RMSD for (**A**) RNA-dependent polymerase with 3-(7-diaminomethyl-naphthalen-2-YL)-propionic acid ethyl ester and (**B**) replication-transcription complex with Thymidine-5′-thiophosphate showing interacting residues.

**Figure 4 molecules-27-04391-f004:**
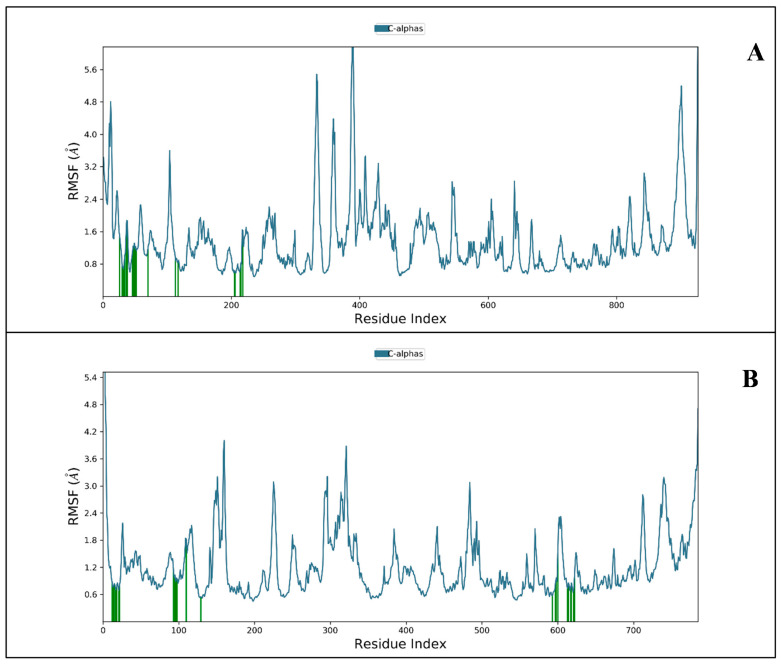
Showing protein-RMSF plot of protein (blue) (**A**) RNA-dependent polymerase (**B**) Replication-transcription, concerning the 3-(7-diaminomethyl-naphthalen-2-YL)-propionic acid ethyl ester and Thymidine-5′-thiophosphate (ligands contact-green).

**Figure 5 molecules-27-04391-f005:**
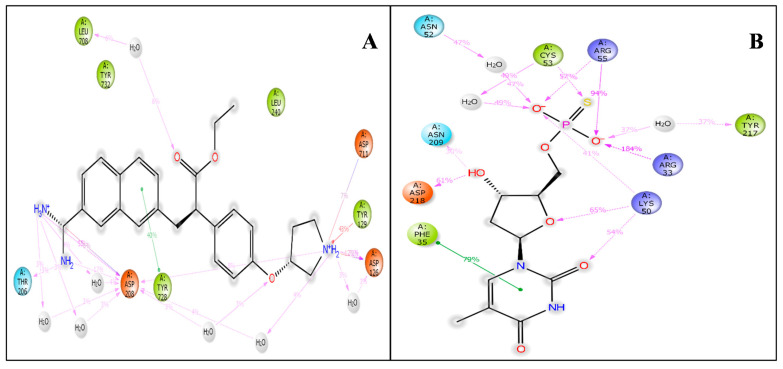
Showing the 2D-Summary of interacting atoms of (**A**) RNA-dependent polymerase with 3-(7-diaminomethyl-naphthalen-2-YL)-propionic acid ethyl ester and (**B**) Replication-transcription with Thymidine-5′-thiophosphate.

**Figure 6 molecules-27-04391-f006:**
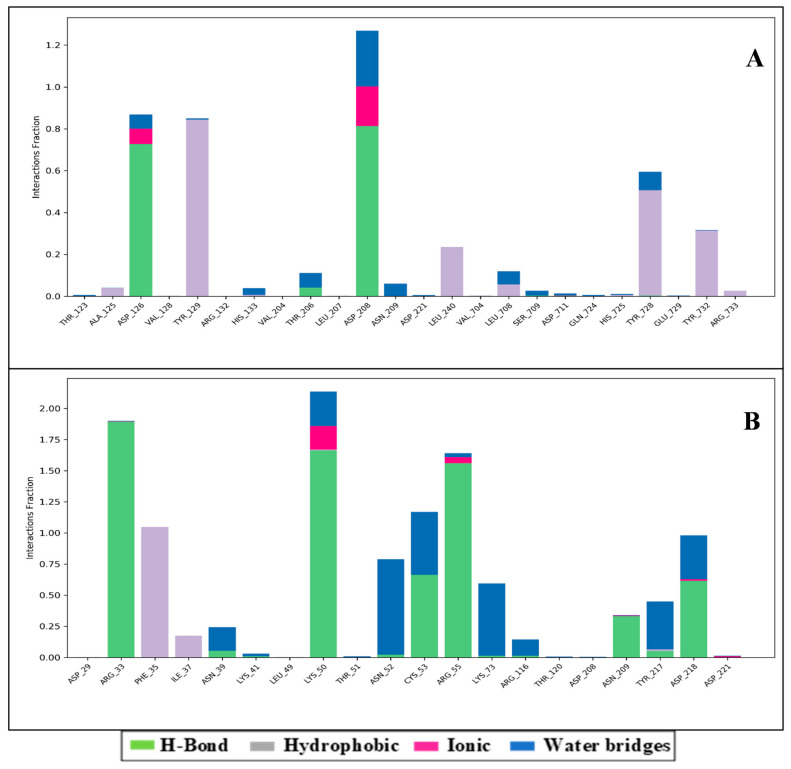
Showing the count of interactions in histogram form for (**A**) RNA-dependent polymerase with 3-(7-diaminomethyl-naphthalen-2-YL)-propionic acid ethyl ester and (**B**) replication-transcription with Thymidine-5′-thiophosphate.

**Table 1 molecules-27-04391-t001:** The Docking Score and binding free energy of the 3-(7-diaminomethyl-naphthalen-2-YL)-propionic acid ethyl ester and Thymidine-5′-thiophosphate) with the Drug Bank ID.

S. No.	Drug Bank ID	Protein Name	Docking Score	MMGBSA dG Bind	Rotatable Bonds	Ligand Efficiency Sa	Ligand Efficiency Ln	Evdw	Ecoul
1	DB07639	RNA dependent-polymerase	−8.781	45.78	9	−1.955	−9.34	−33	−35.6
2	DB08432	Replication transcription	−8.582	55.492	5	−1.042	−3.609	−27.6	−43.4

**Table 2 molecules-27-04391-t002:** The absorption, distribution, metabolism, excretion, and toxicity of the DB07639 and DB08432.

Title	Normal Values	DB07639	DB08432	Title	Normal Values	DB07639	DB08432
#acid	0–1	0	2	IP(eV)	7.9–10.5	0	0
#amide	0–1	0	0	Jm	N/A	0	0.003
#amidine	0	0	0	mol MW	130.0–725.0	433.549	338.271
#amine	0–1	3	0	%HumanOralAbsorption	>80% is high, <25% is poor	50.112	42.465
#in34	N/A	0	0	PISA	0.0–450.0	260.165	29.219
#in56	N/A	21	11	PSA	7.0–200.0	108.278	153.962
#metab	1–8	7	3	QPlogBB	−3.0–1.2	−1.188	−1.776
#NandO	2–15	6	9	QPlogHERG	concern below −5	−8.495	−0.274
#noncon	N/A	4	4	QPlogKhsa	−1.5–1.5	0.55	−0.909
#nonHatm	N/A	32	21	QPlogKp	−8.0–−1.0	−7.049	−5.278
#ringatoms	N/A	21	11	QPlogPC16	4.0–18.0	15.966	10.053
#rotor	0–15	10	6	QPlogPo/w	−2.0–6.5	2.434	1.062
#rtvFG	0–2	1	1	QPlogPoct	8.0–35.0	26.265	20.346
#stars	0–5	0	0	QPlogPw	4.0–45.0	15.875	15.661
accptHB	2.0–20.0	6.25	8.4	QPlogS	−6.5–0.5	−2.394	−2.741
ACxDN^.5/SA	0.0–0.05	0.0176843	0.0317999	QPPCaco	<25 poor, >500 great	3.15	3.31
CIQPlogS	−6.5–0.5	−2.997	−3.155	QPPMDCK	<25 poor, >500 great	1.324	4.15
CNS	−2 (inactive), +2 (active)	−2	−2	QPpolrz	13.0–70.0	47.608	26.903
dip^2/V	0.0–0.13	0	0	RuleOfFive	maximum is 4	0	0
dipole	1.0–12.5	0	0	RuleOfThree	maximum is 3	2	1
donorHB	0.0–6.0	5	4	SAamideO	0.0–35.0	0	0
EA(eV)	−0.9–1.7	0	0	SAfluorine	0.0–100.0	0	0
FISA	7.0–330.0	178.038	240.775	SASA	300.0–1000.0	790.275	528.304
FOSA	0.0–750.0	352.072	186.095	volume	500.0–2000.0	1441.735	915.318
glob	0.75–0.95	0.7809869	0.8629708	WPSA	0.0–175.0	0	72.214
HumanOralAbsorption	N/A	2	2

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
