# Peer review of "In-Silico Screening and Molecular Dynamics Simulation of Drug Bank Experimental Compounds against SARS-CoV-2"

_molecules, 2022, doi:10.3390/molecules27144391_

Round 1
Reviewer 1 Report
Its good work presented by well known team of researchers. I have read manuscript carefully and I think it should just improved in:
1-Abstract should be more concise and specific.
2- Latest references are required to do comparison with presented results.
Author Response
Dear reviewer, we thank you for your kind response and for carefully reading the manuscript. We have improvised the abstract's content and extended the discussion in your suggested direction.

Reviewer 2 Report
Alturki et al, submitted a manuscript, "In-Silico Screening and Molecular Dynamics Simulation of Drug Bank Experimental Compounds Against SARS-CoV-2 ," which presented a virtual screening, ADMET studies.
The paper is well-written in a systematic way.
However, only a point that needs to address in the paper is related to the docking experiment.
-How was the docking experiment verified?
Usually, researchers try to do it with self-docking, where they dock the co-crystal ligand back in the active site of the protein and then compare the docked conformation against its co-crystal binding conformation, which is generally defined in RMSD (Angs.) (for Information: 10.1016/j.jsps.2018.01.017)
This manuscript has elements that make it fit for consideration in the current journal.
-The paper is about the virtual screening of experimental drugs using three different methods (three algorithms). Using three differential algorithms for virtual screening measures the robustness of the adopted methodology as computational methods are flawed in producing the pseudo-positive results; Therefore using more than 2 methods is often advisable. The author further used molecular dynamics (MD) to study the identified inhibitor-target complex where a 100 nanoseconds SPC water model-based simulation was used to analyze the MD simulation trajectories
-The key strength of the paper is the organized way of methodology. Three methods were used for docking, ADMET analysis, followed by molecular dynamics. The authors used docking followed by molecular dynamics and, study the fluctuation of the inhibitor-target complex, which is often used to verify the accuracy of the adopted docking methods. Next, Figure 1 which is a graphical abstract provides a generalized idea of the paper and helps the reader to identify the important sections of the paper.
-There are other minor issues in my opinion should be further addressed:
(a) The authors need to improve their choice of words in some sentences of the paper. For Example, on Page 1 line 30 "both drugs performed flawlessly" should be written in a more scientific way.
Some of the sentences in a manuscript are written in a complex way, authors need to split them into simpler sentences.
The IUPAC naming of inhibitors should be uniform, such as "YL" is inappropriate and should be in small alphabets as the author wrote " 3-(7-diaminomethyl-naphtha- 164 len-2-YL)-propionic acid ethyl ester" in the manuscript.
(b) The choice of target "Respiratory syndrome-coronavirus-2 RNA-dependent RNA polymerase". Authors need to provide an explanation justifying the significance of their choice of target over other available Covid-19 Targets. Secondly, if there is any peculiar structural information related to the active site that was considered for this target, and prompted the authors to choose virtual screening? say example, researchers usually draw the information from the previously identified inhibitor shape or similar binding confirmation for the targets or other structural descriptors.
Author Response
The paper is well-written in a systematic way. However, only a point that needs to address in the paper is related to the docking experiment.
- How was the docking experiment verified? Usually, researchers try to do it with self-docking, where they dock the co-crystal ligand back in the active site of the protein and then compare the docked conformation against its co-crystal binding conformation, which is generally defined in RMSD (Angs.) (for Information: 1016/j.jsps.2018.01.017)
Authors' response:
Dear reviewer, we thank you for your careful reviewing the manuscript. We validated the docking setting with re-docking, and we also analysed the data with the MM/GBSA analysis and with both means, we got the satisfying results, then moved to the molecular dynamics simulation analysis.
- This manuscript has elements that make it fit for consideration in the current journal. The paper is about the virtual screening of experimental drugs using three different methods (three algorithms). Using three differential algorithms for virtual screening measures the robustness of the adopted methodology as computational methods are flawed in producing the pseudo-positive results; Therefore, using more than 2 methods is often advisable. The author further used molecular dynamics (MD) to study the identified inhibitor-target complex where a 100 nanoseconds SPC water model-based simulation was used to analyse the MD simulation trajectories.
Authors' response:
Dear reviewer, we thank you for raising an excellent comment to make us improvise the content. Anyhow all the robust multi-algorithm method was used to expel the compounds that were not properly fitted to the pocket for all the protein cases. And the MD simulation in the SPC water model is the well-accepted model that we have chosen and got an outstanding result.
- The key strength of the paper is the organised way of methodology. Three methods were used for docking, ADMET analysis, followed by molecular dynamics. The authors used docking followed by molecular dynamics and study the fluctuation of the inhibitor-target complex, which is often used to verify the accuracy of the adopted docking methods. Next, Figure 1 which is a graphical abstract provides a generalised idea of the paper and helps the reader to identify the important sections of the paper.
Authors' response:
Dear reviewer, we thank you for your endorsing our efforts to make the manuscript well readable.
-There are other minor issues in my opinion should be further addressed:
- The authors need to improve their choice of words in some sentences of the paper. For Example, on Page 1 line 30 "both drugs performed flawlessly" should be written in a more scientific way. Some of the sentences in a manuscript are written in a complex way, authors need to split them into simpler sentences. The IUPAC naming of inhibitors should be uniform, such as "YL" is inappropriate and should be in small alphabets as the author wrote " 3-(7-diaminomethyl-naphtha- 164 len-2-YL)-propionic acid ethyl ester" in the manuscript.
Authors' response:
Dear reviewer, we thank you for reading the complete manuscript with full enthusiasm and providing an extensive comment. We have improvised the manuscript after your suggestions. Please have a look in this link https://pubchem.ncbi.nlm.nih.gov/compound/46937085
- The choice of target "Respiratory syndrome-coronavirus-2 RNA-dependent RNA polymerase". Authors need to provide an explanation justifying the significance of their choice of target over other available Covid-19 Targets. Secondly, if there is any peculiar structural information related to the active site that was considered for this target and prompted the authors to choose virtual screening? say example, researchers usually draw the information from the previously identified inhibitor shape or similar binding confirmation for the targets or other structural descriptors.
Authors' response:
Dear reviewer, we have described the protein and the database we have taken for the screening in the introduction and discussion section. Anyhow, we have tried some more to put it in better explanatory shape.

Reviewer 3 Report
In-silico screening and molecular dynamics simulation of Drug Bank experimental compounds against SARS-CoV-2
In this study, they have screened the vast library of experimental drugs of Drug Bank with Schrodinger's maestro by using three algorithms, high-throughput virtual screening (HTVS), standard precision, and extra precise docking followed by Molecular Mechanics/Generalized Born Surface Area (MMGBSA). We have identified 3-(7-diaminomethyl-naphthalen-2-YL)-propionic acid ethyl ester and Thymidine-5'-thiophosphate as potent inhibitors against the SARS-CoV-2, and both drugs performed flawlessly and showed stability during the 100ns molecular dynamics simulation. Both the drugs are among the category of small molecules and have an acceptable range of ADME properties.
Abstracs
It is necessary to mention which compound(s) are interesting. Energy free Gibs.
Introduction
Not comment.
Metodology.
Active site calculation and Glide Grid generation: The coordinates of the center and the size of the box used are missing, in addition to the number of poses requested program.
Results and discussion
What are the 691 docked ligands?
Validation of active sites is missing.
Conclusions
Not comment.
Author Response
In-silico screening and molecular dynamics simulation of Drug Bank experimental compounds against SARS-CoV-2
In this study, they have screened the vast library of experimental drugs of Drug Bank with Schrodinger's maestro by using three algorithms, high-throughput virtual screening (HTVS), standard precision, and extra precise docking followed by Molecular Mechanics/Generalized Born Surface Area (MMGBSA). We have identified 3-(7-diaminomethyl-naphthalen-2-YL)-propionic acid ethyl ester and Thymidine-5'-thiophosphate as potent inhibitors against the SARS-CoV-2, and both drugs performed flawlessly and showed stability during the 100ns molecular dynamics simulation. Both the drugs are among the category of small molecules and have an acceptable range of ADME properties.
Abstract: It is necessary to mention which compound(s) are interesting. Energy-free Gibs.
Authors' response: Dear reviewer, we thank you for your for reading our manuscript carefully, we have improvised the content of the abstract.
Introduction: Not comment.
Methodology: Active site calculation and Glide Grid generation: The coordinates of the centre and the size of the box used are missing, in addition to the number of poses requested program.
Authors' response: Dear reviewer, it won't provide any idea while providing the data regarding the grid box size as the coordinates can differ program to program. Anyhow it can be understood better after looking into the docking results; we have provided which residue followed by residue number that are interacting with the ligand.
Results and discussion: What are the 691 docked ligands? Validation of active sites is missing.
Authors' response: Dear reviewer, providing the list of the ligand produced after the initial screening is not useful for future readers. That's why we have not provided it to the journal.
Conclusions: Not comment.
Authors' response: Dear reviewer, we thank you for carefully reading our manuscript and providing us with a chance to improvise the content to make a proof for future readers.